# Group discussions improve reliability and validity of rated categories based on qualitative data from systematic review

Jutta Beher[1,2]*, Eric Treml[1,3¤], Brendan Wintle[4]

1 School of BioSciences, University of Melbourne, Parkville, Victoria, Australia, 2 International Institute of Systems Analysis, Laxenburg, Austria, 3 School of Life and Environmental Sciences, Centre for Marine Science, Deakin University, Geelong, Victoria, Australia, 4 Melbourne Biodiversity Institute and School of Agriculture, Food and Ecosystem Sciences, University of Melbourne, Parkville, Victoria, Australia

¤ Australian Institute of Marine Science (AIMS) and UWA Oceans Institute, The University of Western Australia, Western Australia, Australia

* beherj@student.unimelb.edu.au, beher@iiasa.ac.at, jutta.beher@gmail.com

## Abstract

The number of literature reviews in the fields of ecology and conservation has increased dramatically in recent years. Scientists conduct systematic literature reviews with the aim of drawing conclusions based on the content of a representative sample of publications. This requires subjective judgments on qualitative content, including interpretations and deductions. However, subjective judgments can differ substantially even between highly trained experts that are faced with the same evidence. Because classification of content into codes by one individual rater is prone to subjectivity and error, general guidelines recommend checking the produced data for consistency and reliability. Metrics on agreement between multiple people exist to assess the rate of agreement (consistency). These metrics do not account for mistakes or allow for their correction, while group discussions about codes that have been derived from classification of qualitative data have shown to improve reliability and accuracy. Here, we describe a pragmatic approach to reliability testing that gives insights into the error rate of multiple raters. Five independent raters rated and discussed categories for 23 variables within 21 peer-reviewed publications on conservation management plans. Mistakes, including overlooking information in the text, were the most common source of disagreement, followed by differences in interpretation and ambiguity around categories. Discussions could resolve most differences in ratings. We recommend our approach as a significant improvement on current review and synthesis approaches that lack assessment of misclassification.

## Introduction

The number of reviews in the fields of ecology and conservation has increased dramatically in recent years. Reviews take science to a meta-level that is needed

**Data availability statement:** Data and code is uploaded in figshare: https://doi.org/10.6084/m9.figshare.26889553.v1

**Funding:** John Hodgsons Scholarship was awarded from University of Melbourne ($6400 AUD) to realize the interrater reliability experiment and the Research Training Program Scholarship Australia was in general awarded to conduct the work within a PhD thesis. The funders had no role in study design, data collection and analysis, decision to publish, or preparation of the manuscript.

**Competing interests:** The authors have declared that no competing interests exist.

to transcend the focus and unique circumstances of individual studies, enabling generality. When researchers conduct a review, they search the published literature with a specific question in mind and often use a rubric or filter to find publications that contain relevant content for the question they want to answer. General guidelines for conducting reviews and other meta-analyses aim to facilitate unbiased and repeatable results and include recommendations for asking a second and sometimes subsequent researcher(s) to go through the same process of selecting or interpreting texts using a subsample of included publications and compare the judgments that were being made [1].

A standard part of the methodology when conducting a review is to extract data from the existing literature. While systematic and other structured reviews often classify content into categories, which is most often done by judging content based on a coding scheme, narrative reviews make judgments on content in a rather informal way, possibly drawing more on subjective intuition or interpretation without a standardized protocol. Unfortunately, making judgments about content is not a trivial task, and misclassification can easily occur [2,3]. If error or bias are not detected during peer review, the impact on the field of science could be substantial and long lasting, as even retracted studies continue to collect citations [4,5]. The citation report from Reuters Web of Science for systematic or literature reviews in the field of ecology and conservation shows a steady increase of reviews with an average citation rate of 76 and an h-index of 326 as of June 2021, restricted to reviews and journals with focus on biology and ecology and based on the search terms "systematic review" OR "literature review" AND "ecology" OR "conservation". As reviews are rarely replicated and are, on average, more than twice as often cited as other publications, it is important to ensure the reliability and validity of the coding categories used. Unfortunately, this is not common practice: for example, only three in the 26 most cited and none of the 24 most recent publications that we found in our Web of Science search reported completion of any reliability checks or even stated the general importance of doing so.

Content analysis differentiates between three types of content that differ in the amount of interpretation a reader must do. These types are manifest (coding based on pure detection), latent pattern (coding based on detection plus additional cues) and projective content (coding that requires deduction) [6]. Classification that goes beyond manifest content brings subjective judgments into the process and, hence, uncertainty. Processes that include subjective judgment in addition to pure detection require validation and need to be reliable because of the risk of errors being made during interpretation of different cues in the text. When additional cognitive tasks beyond detection of words are involved in a classification task, for example, interpreting the context in which a word is used and making a judgment based on this interpretation if a specific condition is met, additional possible sources of mistakes or misinterpretation are introduced. General recommendations for reliability tests suggest the use of multiple raters in the process of coding in order to produce reliable data [7–9]. Information on agreement is commonly used during a testing phase to identify problems like unclear categories in the coding scheme and, based on parallel coding of a smaller subset, as an indicator for the reliability of the whole data set.

Multiple metrics [10,11] have been developed to assess rate of agreement between raters as a proxy for reliability. Although reliability is a prerequisite for validity, reliability metrics cannot inform about the validity of tested data [10]. There is a difference between how often people produce the same code (reliability) and how often raters give the best possible answer compared to the actual content of the source (validity). Therefore, agreement that is based on misclassification can increase reliability but at the same time decrease validity and, in the worst case, will make people trust data that does not reflect the actual content of the coded texts.

Unfortunately, testing for validity is difficult. While an objective standard exists for manifest content, against which any rating can be compared (for example, presence/absence like counting the frequency of a word, or testing for a substance in a blood sample), an objective standard in the form of an uncontroversial agreement on interpretation does not exist for latent pattern or projective content and needs to be constructed (for example, the use of a word in a specific context, or the contribution of a detected substance to a health problem). The construction of any standard for latent pattern and projective content requires a certain amount of human judgment, which is inherently subjective, and the resulting standard can, therefore, never be fully objective. The practice to designate the main rater as a point of reference is common [12] but has been questioned because assumed expertise has often been shown to be unjustified when tested [13]. Therefore, a thorough, rigorous process for accounting for subjectivity and error is needed.

The situation that an objectively perceived correct interpretation is unknown and all that can be used is a well-informed subjective judgment is very similar to the context of expert judgment [14,15] and expert elicitation [16]. Therefore, measures that are used to improve data quality in these areas of research, particularly the use of group discussions, seem to be promising tools for testing the validity of coded data in content analysis.

The positive impact of group discussion on agreement between independent raters has been described in the field of medicine [17]. More recent research on expert judgment and elicitation has shown that the reliability and accuracy of judgments usually benefit from an exchange of thoughts and assumptions of individual raters in a group setting [14,18–22]. Even though many of these examples focus on estimation of quantitative data in natural sciences and risk analysis, similar recommendations can also be found in the social sciences regarding qualitative data [12], specifically the importance of discussion during rating exercises when there is *"great sensitivity not only to obvious meanings but also more subtle meanings, and where coders have different levels of knowledge in this regard"* [7]. In the context of business decisions, discussions have been identified as more important than rigorous process [23]. In addition to biases, which represent directional errors, errors that appear as unwanted variability in judgment of people who must choose between options have been termed 'noise' and are often surprisingly high across many professions despite extensive education and training [24]. Here, we consider *mistakes* as errors that create noise and use the term *error rate* when referring to the frequency of these mistakes.

Reading scientific publications is intellectually demanding because their comprehension requires knowledge of the used terminology and concepts in the context of the relevant discipline. Knowledge held by individuals will be similar across scientists from a particular field but will differ in breadth, depth, nuances, and school of thought. In addition to the differences in expertise and the ever-present chance of making simple mistakes, scientists are subject to different kinds of biases when making judgments, both on their own and when in groups. Protocols have been developed that enable people to make judgments with as little bias as possible [19]. To capture the widest possible range of knowledge and subjective judgment, it is important to give individuals the chance to make initial judgments on content on their own. Independent judgments protect from biases introduced through human interaction, such as groupthink and dominance. However, to mitigate biases that matter in individual thought processes, such as availability bias or confirmation bias, feedback and discussion of assumptions and evidence are required.

By facilitating a group discussion after each rater has completed a coding task individually, individual and group relevant biases can be addressed, while unintended disagreement can be corrected when the raters themselves believe they made a mistake and only true disagreement based on convictions is retained. Where misclassification has occurred due

to simple mistakes, it is likely that raters, if given the chance to correct their mistakes, will be able to resolve some disagreements, while new disagreements might be discovered. At the same time, insights into the error rate of individual raters can be gained by quantifying how often raters change their coding decisions, for each category within all rated texts, after discussion. The resulting data would likely be more reliable and more accurate than without such an assessment of evidence for produced codes.

Building on insights from expert judgment as well as social sciences, we describe a case study that trials a combined assessment of reliability and validity of coded data in three steps: By linking conventional parallel coding (step 1) to a subsequent reflection by the raters on their ratings (step 2), followed by a group discussion (step 3), we are able to measure the rate of change due to errors, or changed beliefs. The analysis of these rates of change in agreement produces metrics that go beyond raw initial agreement rates. These metrics include the percent agreement on categories before and after group discussions and error rates for individual raters and variables. The results allow insights into both reliability and validity of the produced codes.

We demonstrate the application of this protocol using 25 published papers that report on conservation decision processes, coded for a structured literature review on conservation decisions [25]. We demonstrate rater learning by comparing error rates and rates of persistent disagreement using five raters who engaged in individual reading/coding followed by group discussions over a 12-month period.

## Methods

To assess the impact of group discussion on the quality of categorical classification of text, we drew on a published dataset of decision-making processes for conservation management [25]. The study sampled the peer-reviewed literature with several search strings to find published texts on conservation management decisions that were derived through a prioritization process. The first author coded all texts for several variables and respective categories, requiring some degree of interpretation, and provided quantitative summaries of these categories as results. The group discussions were used as a means to validate the quality of the codes, with the author (in the following referred to as "main rater") facilitating the experiment and comparing their own ratings and arguments with a group after independent parallel coding.

### Workflow

25 publications were randomly selected from a pool of 466 publications [25], while being stratified across subgroups based on citation rate to include publications with highly cited and rarely cited publications in the experiment. Five raters coded in parallel categories for 23 variables (Table 1) using the provided coding scheme (Supplementary material S1A and S1B Table) and participated afterwards in a group discussion with the option to revise their codes following conversation. All raters were students of environmental sciences with a focus on conservation. Three raters were in the mid or end phase of their Bachelor's degree, and one rater was in the final stages of his Master's degree. Available funds were used as a primary stopping rule and all raters were paid for individual coding and discussions with a standard hourly rate until the money was used up. The moderate sample size of 25 papers is justified as a reasonable sample in light of budget and time constraints [26].

To provide some introduction and training, five publications were used in a pilot study to familiarize raters with the task and to test the clarity of categories within the coding scheme, which resulted in adjustments to the coding scheme where raters reported difficulties (Table S2).

The rest of the experiment consisted of parallel coding of categories for 23 variables for each of 21 publications (one publication overlapped with the pilot) by the same four additional raters. The work was split up into several sessions with a duration of a few hours each to allow raters to remember the details of the rated publications for the group discussion, before rating the next set of publications. Six rounds of coding were needed to work through all publications, each consisting of individual rating of text followed by a group discussion. The first round included one publication, all other rounds

**Table 1. The 23 variables and the number of possible category codes within them. Combinations of options were coded by entering multiple codes separated by a comma.**

| | Variable | Label | Type | Possible categories (in addition to NA) |
|---|---|---|---|---|
| 1 | Context of decision | implementation | nominal | 2 options |
| 2 | Framework used | framework | nominal | 6 options |
| 3 | Country | country | nominal | 186 options |
| 4 | Continent | continent | nominal | 5 options |
| 5 | Spatial scale | spatial scale | ordinal | 6 options |
| 6 | Type of management | management | nominal | 5 options |
| 7 | Threat type | threat | nominal | 11 options |
| 8 | Threat presence | threat present | nominal | Checkbox (presence/absence) |
| 9 | Species | species system | nominal | 14 options |
| 10 | Realm | realm | nominal | 3 options |
| 11 | Type socio-economic considerations | socioeconomic | nominal | 7 options |
| 12 | Inclusion of socio-economic considerations | Socioeconomic present | nominal | Checkbox (presence/absence) |
| 13,14,15,16 | Type environmental, social, economic and other objectives | environmental obj<br>social obj<br>economic obj<br>other obj | nominal | Each 2 options |
| 17 | Count objectives | count objectives | numerical | 4 options |
| 18 | Count of options for decision | count actions | numerical | 4 options |
| 19 | Tradeoff included | tradeoff | nominal | Checkbox (presence/absence) |
| 20 | Sensitivity analysis presence | sensitivity present | nominal | Checkbox (presence/absence) |
| 21 | Sensitivity analysis type | sensitivity | nominal | 9 options |
| 22 | Cost included | cost | nominal | 4 options |
| 23 | Feasibility included | feasibility | nominal | 2 options |

included 3–5 publications. Before the group discussion, the individual ratings were compared, and a list of disagreements was sent out to raters to assist with their preparation for the discussion, with a first option to correct for obvious misclassification. This measure was introduced because the first discussion took more than three hours for one publication, and this change allowed us to successfully reduce the discussion time to one hour per publication, as revisiting the text and confirming evidence for codes was the largest time sink.

The discussions focused on disagreements on any classification. For each category with different classifications from different raters (for example, classification "invasive species" and "pollution" in variable "threat"), each rater shared their evidence from the text on which their judgment was based, as well as any related assumptions. The raters shared their codes for each category in each paper by reading out aloud to create an overview of the range of entries, confirm agreement and discuss existing disagreements. Raters had a second option to adjust their codes during the discussion. The disagreements that remained after code revision opportunities were regarded as true disagreements.

Group effects like dominance and group-think were counteracted by having a first round of individual rating to collect the full range of individual codes and reminding raters during the ensuing group discussion that agreement was not a necessary outcome. The distinction between unintentional misclassification and genuine disagreements over interpretation was repeatedly explained, and counterarguments and true disagreement were encouraged. The discussion was facilitated to make sure everyone was heard, and three different types of questions were asked:

1) Can you provide evidence from the text to justify your code?

2) Do arguments for different interpretations of text exist?

 a. Is the coding consistent with instructions?

 b. Is the coding consistent with other coding entries (e.g., ticking the box for the presence of socio-economic objectives might mismatch with empty related fields)?

3) Have we already discussed similar codes, and is our argumentation consistent?

All data were edited during discussions in Microsoft Excel. One csv file with all codes before each discussion session and one csv file with adjusted codes after the session was created for processing and analysis in R [27], specifically packages *tidyverse* [28], *reshape* [29], *stringr* [30] and *stringi* [31]. Data and code are publicly available on figshare [https://doi.org/10.6084/m9.figshare.26889553.v1].

Studies in which the outcome is a quantitative measure often summarize agreement with Fleiss' *kappa* (in the following referred to as "kappa"), which is a percent agreement measure which accounts for agreement by random chance, for example, when raters have to decide between two categories that are placed next to each other on a quantitative scale. However, our data do not contain many numerical or ordinal variables, hence random agreement was not likely to occur. In addition, metrics like Cohen's kappa, Fleiss' kappa, Krippendorff's alpha or Scott's pi all make the assumption that coding decisions by raters are made independently [10,11]. Our use of a group discussion in the process would violate this assumption. Because of this, and because our data are predominantly nominal and similar to student assessments [32], we decided to tailor our analysis to the data and use percent agreement to calculate metrics of reliability and validity.

We produced two different types of information: first, percent agreement as a measure for reliability of the coding as a procedure that is subject to individual judgment, and second, the frequency with which raters changed their mind about their initial entries as a measure of how trustworthy the code from individual raters was compared to the other raters, and how trustworthy the codes for individual variables were compared to the other variables. This method aims at reducing disagreement based on error, while retaining true disagreements based on interpretation of the categories or the text itself.

Following McHugh (2012), the average percent agreement between the main rater and other raters was calculated as a measure of reliability, and a proxy for the consistency of codes across raters [33]. The average agreement was calculated by first calculating percent agreement for each variable across all coded texts before and after group discussion in pairs between the main rater and each of the other raters, which resulted in four values of percent agreement. The average of these four values was used as a measure of misclassifications and true disagreement on latent content from qualitative text.

The rate of change in coding after discussion was used as a measure of validity, regarding how often the raters thought their own codes to be accurate after having been exposed to a range of interpretations. A measure of confidence regarding collected data is not new, as, for example, Hanea (2017) and Hemming (2018) both incorporated a measure of confidence of individuals who participate in producing estimated data values [14,20]. They asked people to estimate four values, data best estimate, bounds around the estimate, and a value for a self-assessed level of confidence. Their measure of confidence gives information about the intrinsic belief of the rater in their own accuracy. In contrast to their intrinsic measure of confidence, we created a more objective measure of confidence that indicates if a rater's data is robust when tested against other raters, based on how often individual raters change their rating. Our measure of confidence gives information on the ability of each rater to produce codes that are unlikely to be changed when ratings and the underlying evidence in the rated text are scrutinized by others. We recorded the frequency of changed ratings after discussion for all raters across all variables. The resulting set of metrics were (see calculations in S3):

a. The main rater's average error rate and standard deviation can be used as a measure for the likely validity of the full data set (which is coded by the main rater only)

b. Error rates and standard deviation of individual raters are an indication of the importance of multiple raters by providing insight into between-rater variation in error rates and standard deviations. If most additional raters have a significantly lower error rate compared to the main rater, it can serve as an important warning sign regarding the quality of the data set that has been produced by the main rater.

c. Error rates for the categories within the 23 individual variables, averaged across all raters, can guide interpretation and adjustments of data for further use, such as collapsing categories when error rates are high.

We accounted for two potential sources of uncertainty: (i) frequency of categories and (ii) numbers of additional raters. The relative frequency of variables (*prevalence*) can influence the rate of agreement. The comparison of agreement rates between variables that were frequently encountered, and variables that were rarely encountered during the rating can be difficult [34]. We, therefore, compared the metrics for two-factorial subsets of the data for categories for variables that were coded in more than 50% of the produced codes across all raters (frequent subset) and less than 20% of the produced codes (rare subset). The precision of averaged estimates depends on sample size and number of raters and has been shown to improve with increasing numbers of additional raters [8,35]. We tested for the effect of number of coders by calculating percent agreement for all possible main rater/additional rater pairs and averaged the values for all permutations of one, two, three or four of the additional raters. The acceptable level of agreement depends on the purpose. For important decisions a minimum of 90% is suggested, with 80% tolerable in many other settings [36,37].

Recommended thresholds for different agreement rates to pass a reliability test are given in the literature, for example, 60% for kappa [1] or 80% for Krippendorff's alpha [38]. However, generally accepted methods to derive context-relevant acceptable thresholds that warrant drawing conclusions from the data have not been established, and the choice of a threshold is always somewhat arbitrary [9,39]. Here, we consider a percent agreement of > 80% as acceptable.

## Results

We found that group discussion led to a clear increase in percent agreement for all variables (Fig 1) and, by implication, a higher reliability of the coded findings. This pattern was particularly strong for the variables with a high frequency of multiple, combined entries, like multiple threats, species, socio-economic objectives, type of management or type of sensitivity analysis. More than half of the 23 variables showed an initial percent agreement (Step 1) of less than 80%. All variables except *for type of sensitivity* and *species*/*system* passed this threshold after the group discussion (Step 2), with 18 out of 23 variables above 90% agreement and 6 variables 100%. This means almost all disagreement was resolved through discussion (Fig 1). This was not only the case on average but a repeated pattern through all rounds of coding (Fig S4). The improved percent agreement after discussion leads to different conclusions about the reliability of individual variables and categories than the use of kappa or alpha (Table in S5), with 8 categories that would fail the reliability test based on kappa, overlooking that most reasons for disagreement were based on unintended mistakes, and an agreement rate of over 90% after these were detected during discussion.

All raters, including the main rater, changed some codes for at least some categories after the discussions, which allowed the calculation of an error rate for raters and variables (Fig 2). These error rates were compared and used as a measure of validity of answers in the full data set. Most variables with a higher average error rate also had a larger standard deviation, indicating that not all raters made mistakes, while the variable "type of sensitivity analysis" showed a high error rate and small standard deviation, indicating that all raters made many mistakes. The main author kept, on average, almost 90% of their codes unchanged, with a lower average error rate and smaller standard deviation than the additional raters, indicating good quality of coding.

Our sensitivity analysis did not find a clear effect of the frequency of different variables (Fig 2). We found a clear effect of the number of additional coders on the rate of agreement when calculating percent agreement across all possible combinations of raters, while calculating kappa across the same combinations of raters did not show any impact of the

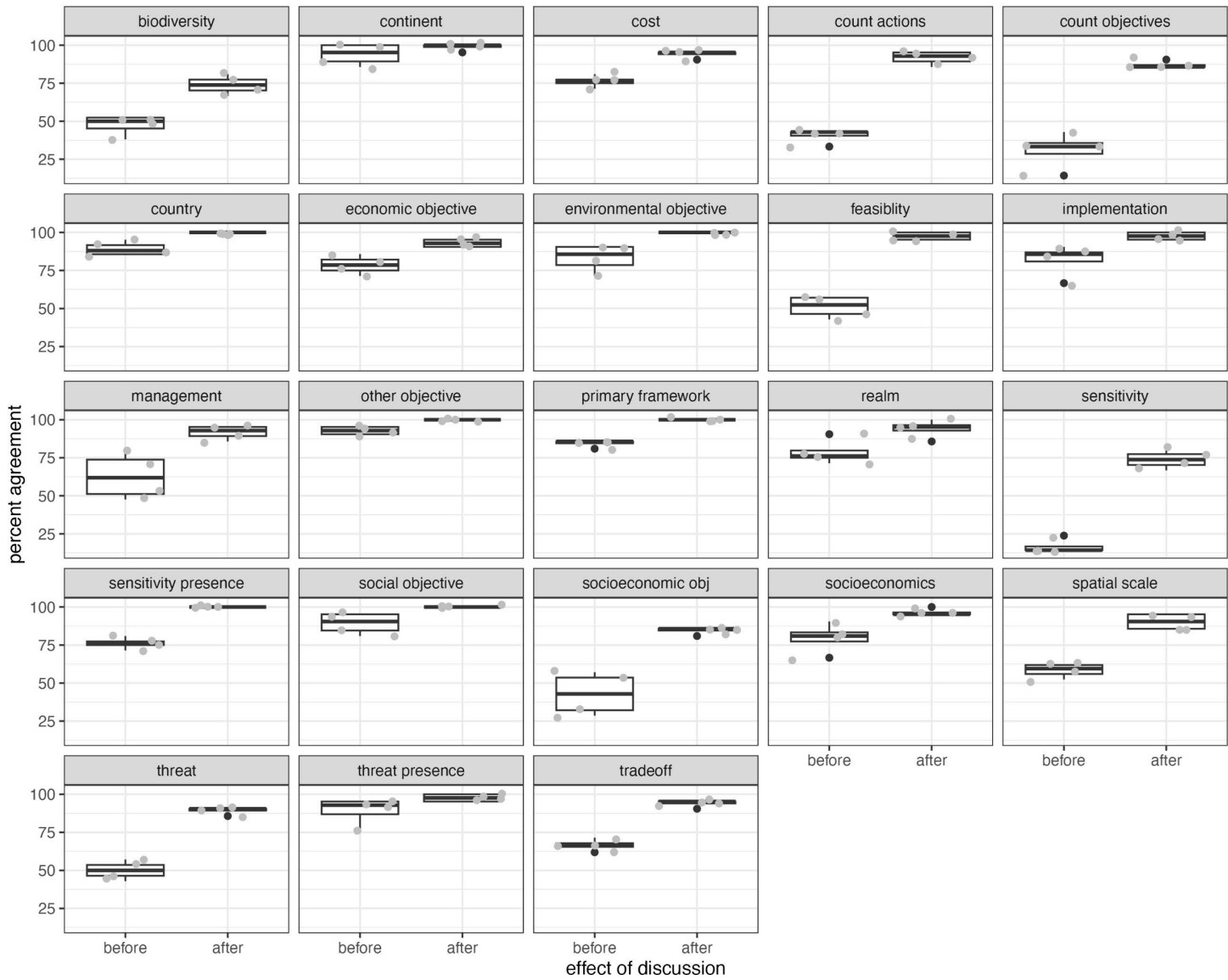

**Fig 1. Percent agreement between the average group of four raters and the main rater for 23 variables before and after group discussions for 21 publications after the pilot.** The effect of group discussion to remove misclassification and retain true disagreements is clear for all variables. Black dots show outliers, while grey dots display all data via jittering. Note that jittering allows to see all points by pushing them apart, in some cases beyond the accurately located outliers in black.

size of the groups except for increased variance, reflecting the larger number of possible rater combinations to compare to each other (Fig 3 - 5). We also did not find any indication of an improvement in the error rate or rate of agreement over the period of the study.

Most variables would fail the inter-rater-reliability test with the use of kappa, with many values below the recommended threshold of 0.6 (Fig 3), reflecting the low percent agreement rates before discussion (Fig 4). Identification of unintended mistakes through group discussion led to overall higher agreement rates, in clear contrast to results with kappa (Fig 5).

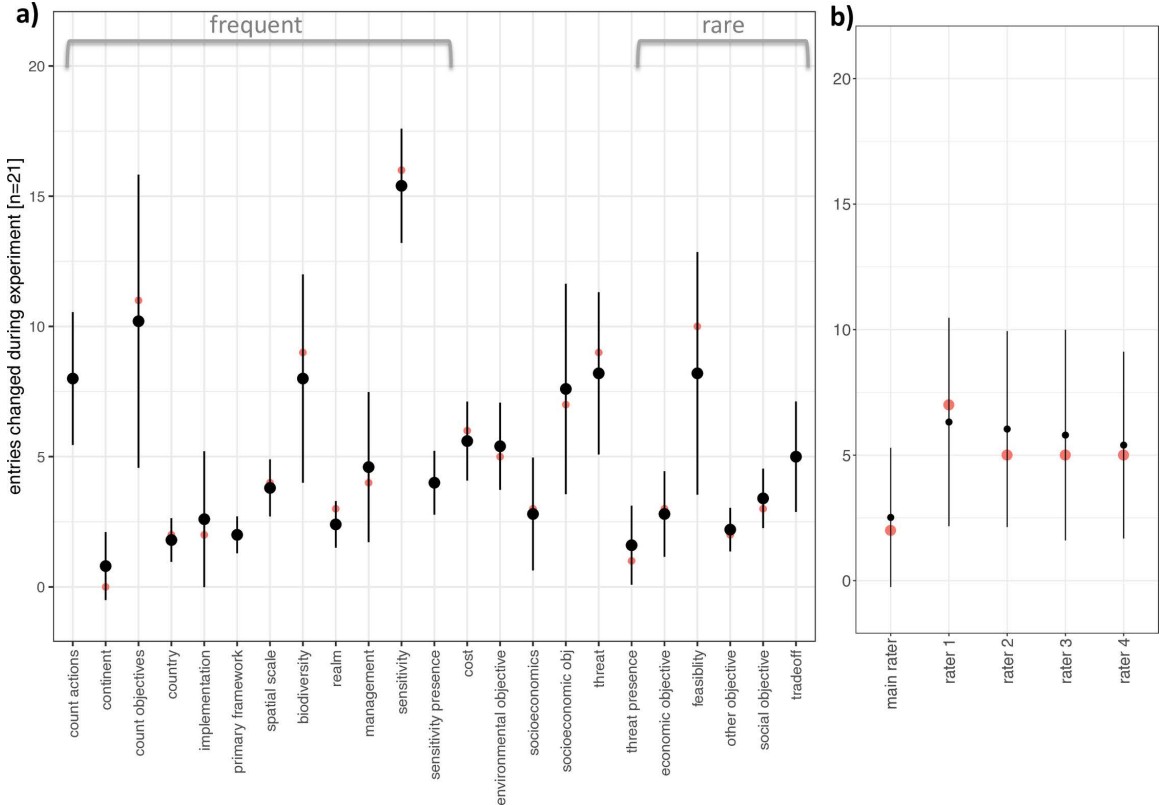

**Fig 2. Average and standard deviation of the frequency of unchanged ratings for each variable across all five raters (a) and each rater across all 21 variables** (b)**.** Medians are shown in red. Variables that were present in more than 50% of codes were considered frequent, and variables present in less than 20% of codes were considered rare.

## Discussion

Humans who make judgments on content make subjective interpretations, mistakes, and have biases, as all these are inherent traits of human thought processes. Our experiment shows that this subjectivity plays a large role, as the overall agreement between raters was low before discussions (Fig 1). However, we also found that agreement dramatically increased following conversation, with an average increase in agreement across all questions of 43% (from 45% to 88% agreement) following discussion. In over 50% of cases, full agreement was achieved following discussion. That is because disagreements arising from mistakes or misunderstandings of text were corrected when raters were given the chance to discuss their judgments and underlying reasoning and evidence with others. Discussion resulted in reduced misclassifications but retention of true disagreements. Our method of pilot assessment and calibration through discussion provides an easy and intuitive way to gain insights into the reliability and validity of literature assessments by reducing unintended variability [24].

What does this tell us about the general process of doing a narrative, structured or systematic literature review? Although guidelines acknowledge the importance of testing reliability [1,9], they do not go into detail about different testing options, and do not provide guidance on how to stem the substantial additional workload and investment that is usually needed to recruit multiple raters. Strategies of rigorous process, testing categories and training raters are suggested to improve the chance of objectively agreeable judgments. However, our results suggest that finding agreement through

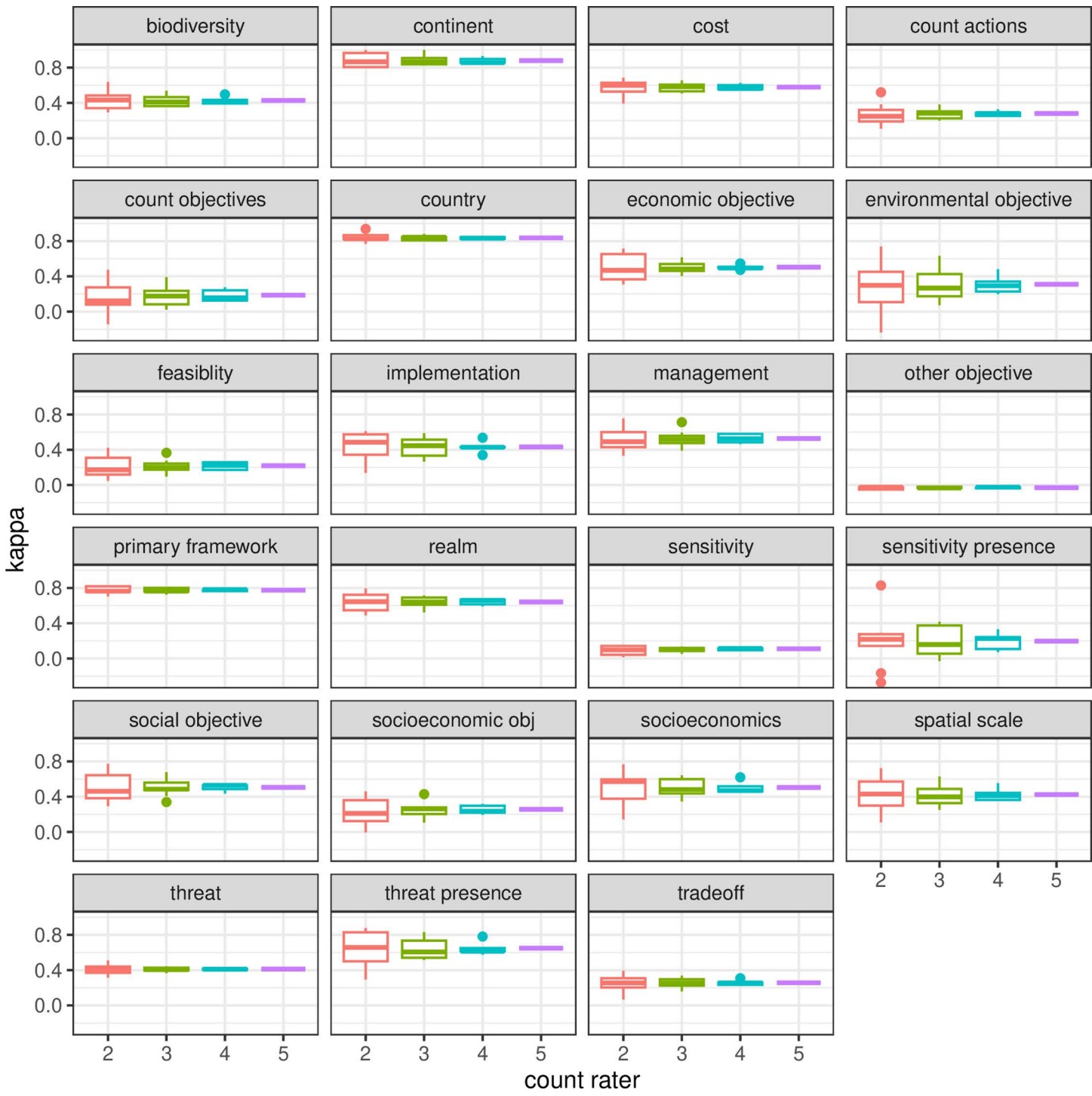

**Fig 3. The calculation of kappa for all possible groups of 2,3,4 or 5 raters did not reveal any impact of group size on pre-discussion agreement.**

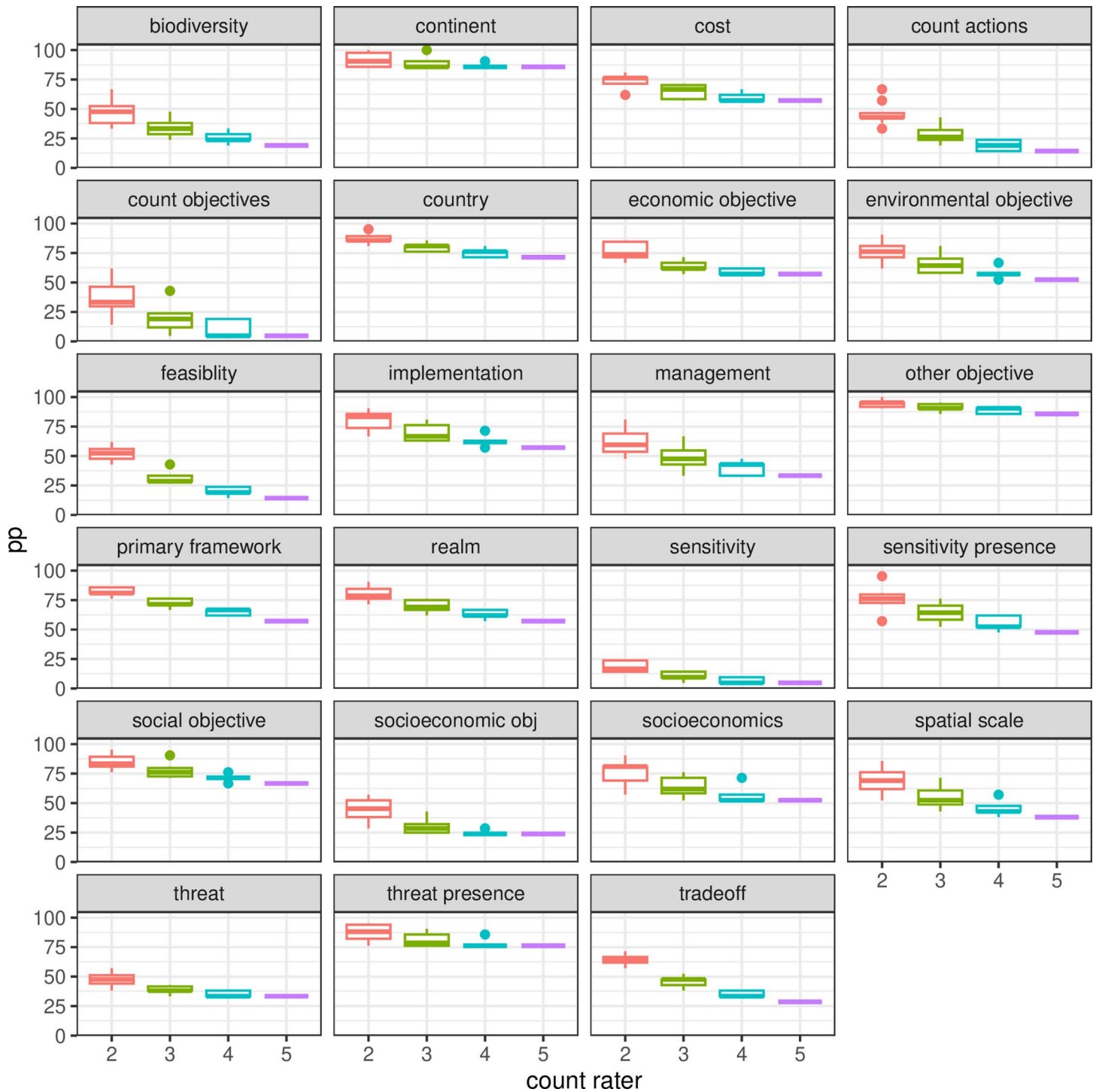

**Fig 4. The calculation of percent agreement for all possible groups of 2,3,4 or 5 raters showed that percent agreement was lower for more raters, which is different to findings for coding of quantitative estimates, where the main across all raters is more likely to converge to the accurate value (Marcoci et al. 2019).**

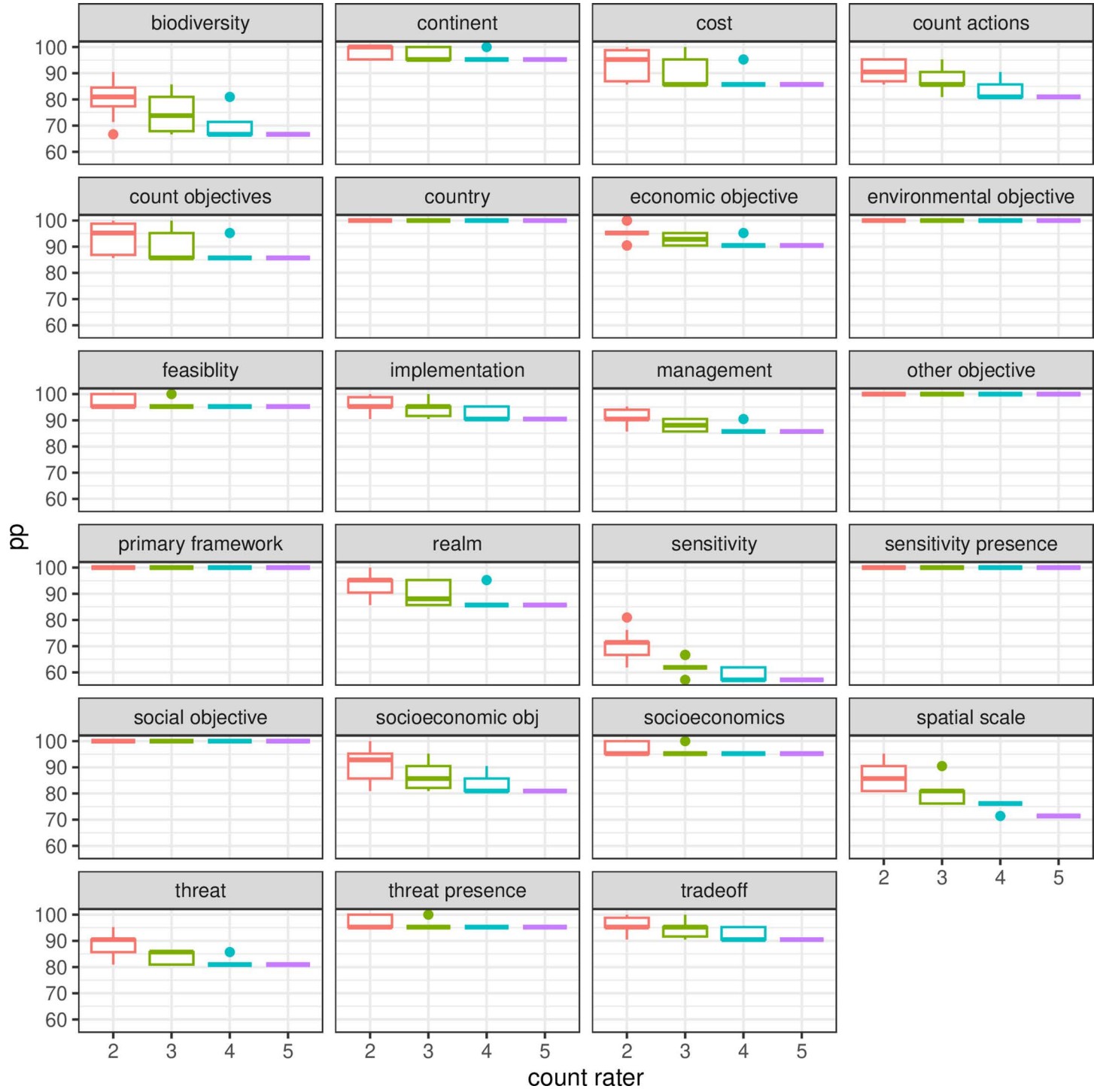

**Fig 5. Percent agreement across all combinations of raters still shows a negative trend for larger groups after the discussion, albeit with overall elevated agreement rates for all variables.**

conversation is very effective in solving problems, including spelling mistakes and oversights, which are unlikely to be overcome with the standard strategies.

Our experiment demonstrates that the opportunity to discuss judgments not only increases agreement between raters but also gives insights into error rates of individuals, which can inform about rater quality. Science relies on narrative, structured and systematic reviews to build broader understanding of patterns and concepts. Scientists who undertake a review need to be aware of the potential large error rate they might have when judging texts, especially when judgment and interpretation of context in addition to the detection of plain presence of specific words is required. Being aware that review processes are inherently subjective processes should encourage scientists to calibrate their own reliability by exposing themselves and their rubrics to the critical feedback of others during the review process. We all strive for accurate insights when doing reviews, but without the help of others, it is not possible to find out if we made mistakes in our judgments.

A reliability-checking process which relies on calculating kappa to see how reliable coding is between raters *but does not engage in discussions to interrogate why raters might have disagreed* will not be able to distinguish unintended mistakes from actual, earnest differences in interpretation. Mistakes create unwanted variability on the individual level, obfuscating the detection of legitimate differences in interpretation. Identifying these mistakes makes it possible to detect and address problems, including ambiguities in the coding scheme or difficulty of the rated content.

We therefore believe our method to be better equipped to assess the overall quality of text-derived qualitative literature analysis than metrics that evaluate initial agreement, such as kappa or alpha, which account for chance [10]. For example, 8 variables that would fail the interrater test by using kappa show over 90% percent agreement rate after discussion, indicating the high rate of unintended mistakes (Table S5). This is particularly the case when ratings are made beyond manifest content (for example, coding based on pure detection of specific words) and are subsequently used for quantitative summaries and synthesis of qualitative information. As this is the case for many literature reviews, and these syntheses are highly cited and influential, the additional information that provides a quality-check is worth obtaining.

There were three different types of causes for disagreement: disagreement by mistake, disagreement due to ambiguity in categories, and disagreement due to ambiguity in the content that was to be classified. We discuss each of these and the implications for reliability and validity in the following.

**Mistakes were the most common reason for disagreements, rather than fundamental disagreements of interpretation** Agreement rates increased drastically after the opportunity to correct mistakes after exchange (Fig 3 and 5). The most frequent underlying causes tended to be forgetting to check tickboxes, entering the wrong code number, and overlooking text. The benefit of discussions to flag mistakes was very clear but time-consuming. We were able to improve time efficiency considerably by building some interdependencies and overlap into the coding scheme. Doing so allowed us to check for inconsistencies and flag errors in raters' answers prior to discussions. For example, a question that asked whether socio-economic objectives were present was somewhat redundant, given that another question asked raters to identify the "*type of socio-economic objective*" present in the study. But the combination made it often possible to detect missing entries, and raters could be given the chance to fill in their gaps and join the discussion with complete codes.

Mistakes were often caused by difficulties in sustaining attention during reading. Coding for variables that required picking up multiple cues in different passages of the text created significant challenges for raters and led to disagreement between raters. For example, there was high agreement between raters about whether sensitivity analysis had occurred in the analyzed texts. However, there was low agreement about what the specific types of sensitivity analysis were. In most cases, multiple descriptions of how uncertainties were explored were mentioned throughout manuscripts, and there were rarely dedicated concise sections in the analyzed texts. Discussions revealed that the most common cause of rater disagreements for this variable seemed to arise from raters simply overlooking cues within the text, and not from lack of clarity in the writing. This could have been caused by a lack of understanding on the part of the raters about the subject matter in the texts, such that they could not discern correct answers from the available cues, or plain mistake by skimming or not focusing enough while reading.

## A second cause of disagreement was an inherent ambiguity of categories

This was the case when coding categories may not have been clear or specific enough such that an unambiguously correct rating exists. Despite inbuilt mitigation via testing codes in a pilot and collapsing problematic categories, some residual ambiguity usually remains. An unfortunate characteristic of classification schemes that are founded on linguistic terminology is an inherent "fuzziness", which results in a lack of clear demarcation between different categories [40], inviting ambiguity. Semantic ambiguity in the available classification options presented to raters can introduce uncertainty and errors in answers they provide. For example, "mangroves" were classified as an individual species, trees, or a type of ecosystem by different raters. In another study, "native woodlands" were classified as forest or ecosystem by different raters, and there were different opinions on whether "unspecified locations in Victoria" were best classified as local or regional spatial scale, as both interpretations seemed plausible [41]. Although the discussion proved helpful in explaining the underlying cause of disagreement in the case of ambiguous categories, it did not help to resolve disagreement in all cases. Identifying for which variables this is an issue can help to refine the rating scheme and interpret the results.

Our pilot phase of the inter-rater-reliability testing had the objective of mitigating existing ambiguity as far as possible. We expected that we would be able to minimize the issue but not remove ambiguity completely, due to the inherent fuzziness of linguistic categories. Our results, therefore, confirm that even though ambiguity in coding schemes can rarely be removed completely, a pilot phase that aims to detect issues can minimize problematic codes.

## Some texts were difficult to rate due to under-specificity in the text itself, requiring interpretation of content

Our findings support evidence that different readers package qualitative content in different ways when tasked with classification, despite overall agreement on broader themes [42].

Some interpretation was easier when raters had advanced knowledge of mathematics, ecology or decision theory, but some cases relied on purely subjective interpretation. For example, one text described several distinct species and taxa, such as waterbirds, in the introduction, but it remained unclear if these species were considered in the case study or just used as a reference for similar studies [43]. The discussion among raters could not clarify if the case study included data on waterbirds or only broader ecotypes because the underlying text was not specific enough, and no one could bring forward clear evidence for one case or the other. In a similar example, disagreement remained over whether "non-target species" in a plan for wild pig control were best classified as fauna or not because, although an example with other animal species was given, it was possible to imagine that some of the control activities would have an impact on vegetation [44].

Discussion and identification of frequent mistakes enabled the identification of problematic categories. In our study, seven of the 23 variables were coded with a high error rate, a low initial agreement rate and a higher standard deviation than other variables (Fig 2). Most of these seven variables (species, count of action, count objectives, type of threat, presence of feasibility, type of sensitivity analysis, and type of socio-economic objective) were more difficult for some raters than for others, although a general underlying cause for the divergence in rating is hard to determine. The exception is the variable "type of sensitivity", which can be identified as the only variable that was difficult for all raters, indicated by the high change of mind and low standard deviation. Collapsing categories can sometimes be the only solution to accurately code such a problematic variable. For example, it was often the case that raters had only detected a subset of the evidence in the texts regarding the variable *type of sensitivity analysis* and realized, once made aware of relevant descriptions in the text, that they had not paid enough attention and needed to add missing categories to their ratings, which led to high agreement after discussion. Another example was the broader agreement that uncertainty had been explored in one of the texts [45], but raters could not settle on one classification for the relevant parts in the text as different interpretations seemed plausible to choose the category of different values of parameters, different models, scenarios for different actions, differences of given scores, or a combination of these methods. The reduction to a presence/absence code when disagreement affects only nuances within a specific variable can serve as a more cautious and reliable replacement, albeit incurring a loss of information.

**The main benefit of comparing different raters' error rates was to be able to make a judgment on the coding quality of the main rater**

The main rater coded a much larger number of texts that could not all be included in a parallel coding stream due to time and cost constraints. But the comparison of error rates gave confidence in the untested codes because, on average, all additional raters had a similar error rate and changed their codes more often than the main author. In the hypothetical case that the main rater has a higher error rate, this information could flag concerns about the quality of the codes. In the case that there is no main rater and different raters code similar fractions of the sampled texts with some overlap, our method can identify raters that perform worse or better than others. This helps to make a judgment on the reliability and quality of the produced codes. If variables that have high error rates in all combinations of raters are detected, categories can be collapsed or dismissed. See S2 for examples of collapsed categories in our study that were identified during the pilot phase.

**High heterogeneity of qualitative content in papers, as was the case in our study, makes it difficult to create the opportunity for training and learning**

Training and learning ahead of rating have been important and well-proven mechanisms to increase accuracy of ratings for quantitative estimations [46]. The rationale that training can prepare sufficiently for rating procedures, and a time- and budget-intensive discussion after rating is therefore not necessary, might be posed as an objection to our proposed method. In the context of quantitative estimates, learning occurs more readily when quantities are repeatedly estimated in a consistent problem context, such as assessments of symptoms of a particular condition in health or assessments based on quantitative ecological or economic data [46,47]. In these settings, multiple rounds of feedback on estimated quantities enable raters to calibrate themselves through learning.

However, there are important differences between quantitative and qualitative codes, especially when content that needs to be rated is very heterogeneous.

While the repeated activity of estimating quantities seems a straightforward method to hone a specific skill, the required concentration to read long texts and pay careful attention to multiple qualitative categories and variables is a very different challenge. Our results have shown that most disagreement stems from a lack of focus and careless mistakes but very rarely from a misunderstanding of key concepts or a lack of knowledge that could be bolstered through training. A decreasing error rate over time would indicate that raters improve in accuracy by doing the same task repeatedly over time. While evidence has been collected to back up training as a tool in quantitative estimates [18,46], we could not detect such a decrease in error rates and, therefore, cannot confirm that raters improved their coding of nominal variables beyond manifest content by coding the same categories repeatedly over time (Fig S4). We also did not find that agreement was impacted by the frequency with which a variable was encountered. We interpret this as further evidence that training exercises with very heterogeneous data might be suboptimal compared to group discussions.

The lack of clear demarcation between classifications might also play a role in explaining why we could not detect any evidence of learning in the form of improved accuracy over time, even though some texts dealt with similar conservation context regarding specific groups of animals or threats. If the heterogeneity of the publications diminished the chance to improve rating through learning, this has wider implications: Any commonly employed preparatory exercises or training before the actual rating will be much less effective in improving the overall quality of the codes than discussions during the rating. This is in line with evidence that outcome-based feedback does not necessarily improve the accuracy of ratings, while group-based judgment for quantitative estimates seems to be more accurate and reliable than individual judgments [46].

Because of the assumed difference in underlying cognitive mechanisms of rating quantities (a skill that can be trained and calibrated) and detecting cues in text (which depends on prolonged concentration and attention to detail), we believe there are also differences in dealing with the collected quantitative and qualitative data. A commonly

used strategy for dealing with divergent rater assessments in expert elicitation and modeling contexts is to compute answer averages [8,13,18,19,48,49]. Unfortunately, qualitative categories do not lend themselves to averaging due to their fuzziness.

Although errors in codes of qualitative categories cannot be smoothed out through aggregation as is the case for quantitative estimates, a non-mathematical smoothing effect might be produced through the sharing of arguments and evidence, which enables raters to adjust their best attempt to choose a fitting variable category code [46]. The difference between quantitative and qualitative content might also explain why we found a decrease in percent agreement with increasing group size, which is the opposite effect of numbers of raters on agreement in Marcoci et al. (2019). While our study investigated latent pattern content (detection of presence in context), Marcoci et al. investigated projective content (assessment/judgment of quality expressed through ordinal scales), which has more similarities to quantitative data and can benefit from mathematical smoothing effects.

### Caveats of this study

**Limited number of assessed publications and qualification of additional raters.** When using the difference in error rates as an indicator of the reliability of the main rater, results will depend on the quality of the additional raters. Additional raters need to have sufficient education to understand the task and the content of the texts, and the necessary rigor for a successful rating experiment. In our case study, three raters were in the mid or end phase of their Bachelor's degree, and one rater was in the final stages of their Master's degree. The main rater was a PhD candidate. It is reasonable to assume that the main rater will likely have spent more time reading and thinking about the question and the coding scheme, as well as being emotionally more invested in a quality result. However, it is also reasonable to assume that more senior scientists that fit the profile of an additional rater for the coding task might be less often available and interested in taking part in such a time-intensive and poorly paid exercise.

The limited number of our sample of 25 coded publications gives evidence of how time- and budget-intensive an orchestrated group discussion is. We used available funds (~AUD 10.000) as a primary stopping rule, and all raters were paid for individual coding and discussions with a standard hourly rate until the money was used up. We, therefore, believe the limited sample size of 25 can be justified in light of budget and time constraints [26].

**Additional aspects of inter-rater-reliability were not explored here but might provide further insights.** The agreement within the group of additional raters relative to the main rater's code can be used as a guide to quality. Variables that show high agreement among the group of additional raters and low agreement between individuals and the main rater could indicate problems, while variables that show high agreement among the group of raters and high agreement with the main rater could be used as a general sign of quality (see Fig S4).

Our study does not address the challenges and possible solutions to coding mistakes and uncertainty for manifest or projective content. Future studies could seek to clarify whether multi-rater discussion and calibration improve the quality of reviews based on such data. We would anticipate that whenever the production of codes involves any type of judgment and prolonged focus, there is a strong potential to reduce mistakes and improve validity of codes through discussions of underlying assumptions and reflection on the strength of evidence provided by raters.

We were not able to examine effects that different media can have on reading and focus, with clear advantages of print media for comprehension and focus [50,51]. While future studies could test the effect of reading medium on agreement rates, our results are an important reminder to any academic that comprehension of texts requires prolonged focus and mistakes that lead to poor comprehension of content can easily happen, even for seemingly obvious things like species, methods and countries included in a study.

**The progress of artificial intelligence and machine learning and its potential for automatization of qualitative judgments of texts.** Since this study was conducted, machine learning and artificial intelligence applications that digest text for specific purposes have accelerated in their development and application. However, generative AI and

chatbots make mistakes, too, and suggest for example citations that do not exist or code that does not work. Such errors are often referred to as "hallucinations". While problems like hallucinations persist and high accuracy at scale required for most academic work has not been achieved yet, programs become progressively better in text-related tasks, especially extraction of information from text [52–55]. Despite the persisting challenges, researchers see an increasing role of AI in research and for reviews in particular [56]. Future studies could compare results across human and electronic raters to assess and monitor if and when these tools will be of the needed quality to apply for contextual classification tasks.

## Conclusion

Our work provides further evidence that extraction of qualitative data from free-flowing text is a demanding task and benefits from the attention of multiple people. Discussion improves agreement when multiple people interpret the same text by reducing mistakes. The positive effects of discussion on rates of agreement and patterns of error rates were clear and consistent across all variables and raters.

In light of our results, we highly recommend including a targeted inter-rater-reliability test in any classification task from text that goes beyond kappa statistics. Some issues of classification are not related to specific coder experience or skill but are inherent in free-flowing text documents, including unclear structure and terminology. Having more than one rater to compare error rates against will improve understanding of the quality of code and improve trust in quantitative and qualitative results that stem from sampling of large numbers of texts. As different types of reviews and summaries of literature are specifically part of doctoral studies, we encourage students to seek out existing additional funding sources, and Universities for providing support, to enable a systematic endorsement and best practice of conducting reviews by elevating such studies from solo enterprise to a quality checked group effort.

## Supporting information

**S1 File. S1A Protocol instructions.** Instructions for raters on objectives of review and data entry.
(PDF)

**S1 Table. S1B Table. Coding scheme cheat sheet.** Full list of classification categories and explanation.
(PDF)

**S2 Table. Changes made after the pilot.** Details on all changes made after the pilot phase to the coding scheme including reasons.
(PDF)

**S2 File. Calculations for figures.**
(PDF)

**S1 Fig. Average agreement across all pairs of raters compared to the main rater.** S1 Fig. Average agreement across all pairs of additional raters compared to the main rater before and after each of the 6 rounds of group discussions. Discussed examples from publications were n = 1 during the first discussion, and between n = 3–5 in all other discussions.
(TIF)

**S3 Table. Comparison of metrics for agreement.** Kappa for individual raters and all raters, alpha for all raters, and percent agreement before and after discussion. The percent agreement after discussion differs drastically for some variables from kappa or alpha.
(PDF)

## Acknowledgments

We thank the four raters, Jessica Keem, Erin Bernadette Thomas, Diego Brizuela Torres and Matthew Paul Whitney, for their participation. Thanks to Dr. Bonnie Wintle for her help in revising the coding scheme and Prof. Fiona Fidler for advice on the design of the experiment and for providing comments on the draft manuscript. Special thanks to Steven Kambouris, who helped immensely by providing advice on interrater reliability and analysis in general and providing extensive comments on the draft manuscript.

## Author contributions

**Conceptualization:** Jutta Beher.

**Data curation:** Jutta Beher.

**Formal analysis:** Jutta Beher.

**Funding acquisition:** Jutta Beher.

**Investigation:** Jutta Beher.

**Methodology:** Jutta Beher.

**Supervision:** Brendan Wintle, Eric Treml.

**Visualization:** Jutta Beher.

**Writing – original draft:** Jutta Beher.

**Writing – review & editing:** Jutta Beher, Brendan Wintle, Eric Treml.

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
