## [Decision Letter · Decision Letter 0]

Mar 23 2025

Dear Dr. Beher,

Thank you for submitting your manuscript to PLOS ONE. After careful consideration, we feel that it has merit but does not fully meet PLOS ONE’s publication criteria as it currently stands. Therefore, we invite you to submit a revised version of the manuscript that addresses the points raised during the review process.

I've received feedback from two reviewers. Both found your manuscript to be interesting, relevant, and well-written. I agree with them. The reviewers had some comments that you should address, but these are mostly minor and should be straightforward to resolve during revision.

We look forward to receiving your revised manuscript.

Kind regards,

Frank H. Koch, PhD

Academic Editor

PLOS ONE

Journal Requirements:

John Hodgsons Scholarship was awarded from University of Melbourne ($6400 AUD) available to realize the interrater reliability experiment and the Research Training Program Scholarship Australia was in general awarded to conduct the work within a PhD thesis

3. Please ensure that you refer to Figure 1a in your text as, if accepted, production will need this reference to link the reader to the figure.

4. Please include a caption for figure 1a.

5. Please include a copy of Table 2 which you refer to in your text on page 9.

Additional Editor Comments :

I've received feedback from two reviewers. Both found your manuscript to be interesting, relevant, and well-written. I agree with them. The reviewers had some comments that you should address, but these are mostly minor and should be straightforward to resolve during revision.

Specific comments:

Line 73: "judgment" instead of "judgement" (for consistency with elsewhere in the text)

Line 231: insert "of" after "consisted"

Line 510: replace "couldn't" with "could not"

Reviewers' comments:

Reviewer's Responses to Questions

**Comments to the Author**

1. Is the manuscript technically sound, and do the data support the conclusions?

Reviewer #1: Yes

Reviewer #2: Yes

2. Has the statistical analysis been performed appropriately and rigorously?

Reviewer #1: I Don't Know

Reviewer #2: Yes

3. Have the authors made all data underlying the findings in their manuscript fully available?

Reviewer #1: Yes

Reviewer #2: Yes

4. Is the manuscript presented in an intelligible fashion and written in standard English?

Reviewer #1: Yes

Reviewer #2: Yes

Reviewer #1: This paper presents an interesting investigation into using group discussions to improve the coding of papers in systematic reviews in conservation. I particularly appreciate the fact that the author paid attention to pragmatic costs and feasibility in their analyses which are important considerations. I don't have much to offer in the way of specific improvements.

My one main comment is that in this day and age, the potential for combining human reviewer efforts (including group discussions) with machine-based reviews seems like it is a critical topic that needs to be addressed in this paper. I realize that the actual study being reported in this paper did not use AI tools and I don't think it would be at all feasible to add them into the mix at this point. But I would expect to see at least a couple of sentences in the discussion talking about the potential of combining human and machine based reviews - perhaps a topic for a future study???

Reviewer #2: This is a really interesting and important paper. It is well written and makes some useful arguments. I am happy to recommend the paper be published if the authors can attend to some minor amendments and suggestions below:

Methods: I agree with the authors on pg 8 lines 178-184 that the information about the systematic review and PRISMA figure can be removed. Instead the authors could provide a short description of the original study and reference it (since it has been published)

Some suggestions:

1. On pg 10, subsection titled Workflow – the authors refer to 25 studies used to illustrate their argument as ‘examples’ – this is a bit confusing – I would suggest they use the terms 25 studies, and coding on 23 categories (were there sub-codes?) and amend all references to ‘examples’ as used above to studies or publications (to make it absolutely clear what they are referring to).

Subsequently the authors refer to ‘case study’ and it is not clear what they are referring to – an ‘example’ or results of a round of coding?

2. Perhaps the author/s could mention who the main rater was – and their expertise that made them the main rater earlier in the paper (it comes very late in the discussion). It would also be useful to mention whether any training was given to the additional raters before the process and whether they all had domain expertise (albeit to varying levels).Although on pg 27 it is mentioned that in case the studies being coded are very heterogeneous, training of coders can be difficult so as to improve coding agreement – nevertheless some initial training needs to have been provided and some mention of that would be useful.

3. Pg 10: line 245 – perhaps the authors would like to clarify whether each time different 3 -5 studies were used in each round of subsequent coding – [it is a little confusing whether all studies were coded only once by coders (in different rounds) or some studies were coded twice or more in different rounds.

4. From the description it appears to me that the group discussion took place only after all the rounds of coding were completed – If that was the case, I am not sure what was the point of doing the coding piecemeal in rounds? It is a bit unclear.

5. Were the authors able to conclude whether there is an optimum number of additional coders – (marginal benefit of additional coders and at what point does it become negative)? It might also be worth speculating whether the marginal cost of an additional coder (time, effort and funds) is worth the marginal benefit?

6. Pg 25: if coding some categories was more difficult for some coders than others – what does it say about the level of domain expertise additional coders must have. Perhaps the authors would like to make some recommendations?

7. Finally, perhaps the authors would like to comment on how might PhD candidates who are conducting a systematic review, as part of their thesis, incorporate (and pay for) additional coders and what impact that might have on whether this would be acceptable. Furthermore, would the implications of this study suggest that systematic reviews should not be a solo enterprise?

Just a minor edit: Pg 8 line 173 – Perhaps the authors would like to remove what seems like an accidental reference to the fact that the paper has directly been lifted from a larger thesis.

**Do you want your identity to be public for this peer review?** For information about this choice, including consent withdrawal, please see our Privacy Policy

Reviewer #1: No

Reviewer #2: **Yes: ** Jyoti Belur

---

## [Author Response · Author response to Decision Letter 1]

2 Apr 2025

PONE-D-24-38272

Group discussions improve reliability and validity of rated categories based on qualitative data from systematic review

PLOS ONE

Please find in the following a detailed response to all suggestions, referring to line numbers in the manuscripts where changes have been made, and edited text in blue.

PLOS ONE Journal Requirements:

and

I have checked the guidelines for text and figure files and made the following changes:

• Updated formatting of list of authors and affiliations to remove addresses and postcodes and use appropriate symbols for corresponding author and equal set of authors and current addresses

• Updated all level 1 headings to be fontsize 18 (in addition to be bold)

• Changed all references to figures from “Figure” to “Fig” in manuscript and Supplementary material

• Transferred all Annexes from main manuscript to individual files and edited Supplementary material section to provide correct naming convention including titles and captions for figures and tables.

• Changed bolded sentences in discussion paragraph to subheading level 2, and edited some sentences to make it work (line 538: added “This was the case” and removed the comma after first part of sentence that become the subheading in line 536).

Thank you for these instructions, I have uploaded all figures and downloaded the improved versions that all passed. I have added the images via the editorial manager.

3. Please ensure that you refer to Figure 1a in your text as, if accepted, production will need this reference to link the reader to the figure.

Following the comment of reviewer 2, the section of text that would have referred to the figure has been deleted

4. Please include a caption for figure 1a.

Please see reply above.

5. Please include a copy of Table 2 which you refer to in your text on page 9.

Please see reply above, the text has been deleted.

Many thanks for the instructions, I have updated all supplementary material information according to the link and provided headlines and captions in the manuscript as well as individual files for each as uploaded document. I have updated the in-text references accordingly in marked up changes.

I have double checked all references. I noticed that I had one publication in the reference list with 2 numbers (46 & 48, and 7&23) and removed the second entry for each. A few references had some spelling errors and capitalization, which I changed. References for R packages were updated (line 918 (28), and line 922 (30), and one reference added due to new text (Beher et al 2024 in the methods, #25).

Additional Editor Comments :

I've received feedback from two reviewers. Both found your manuscript to be interesting, relevant, and well-written. I agree with them. The reviewers had some comments that you should address, but these are mostly minor and should be straightforward to resolve during revision.

Specific comments:

Line 73: "judgment" instead of "judgement" (for consistency with elsewhere in the text)

Thank you for catching this inconsistency, we have corrected the spelling (now line 74).

Line 231: insert "of" after "consisted"

Thank you for catching this omission, we have added the word (now line 250).

Line 510: replace "couldn't" with "could not"

Thank you for the suggestion, the change was made (now line 533).

Reviewers' comments:

Reviewer's Responses to Questions

Comments to the Author

1. Is the manuscript technically sound, and do the data support the conclusions?

Reviewer #1: Yes

Reviewer #2: Yes

2. Has the statistical analysis been performed appropriately and rigorously?

Reviewer #1: I Don't Know

Reviewer #2: Yes

3. Have the authors made all data underlying the findings in their manuscript fully available?

Reviewer #1: Yes

Reviewer #2: Yes

4. Is the manuscript presented in an intelligible fashion and written in standard English?

Reviewer #1: Yes

Reviewer #2: Yes

5. Review Comments to the Author

Reviewer #1: This paper presents an interesting investigation into using group discussions to improve the coding of papers in systematic reviews in conservation. I particularly appreciate the fact that the author paid attention to pragmatic costs and feasibility in their analyses which are important considerations. I don't have much to offer in the way of specific improvements.

My one main comment is that in this day and age, the potential for combining human reviewer efforts (including group discussions) with machine-based reviews seems like it is a critical topic that needs to be addressed in this paper. I realize that the actual study being reported in this paper did not use AI tools and I don't think it would be at all feasible to add them into the mix at this point. But I would expect to see at least a couple of sentences in the discussion talking about the potential of combining human and machine based reviews - perhaps a topic for a future study???

Thank you for bringing up this important topic – we agree that since the time of the work, AI has made incredible progress and applications increase in unseen speed and numbers. While we are not experts in the field, we have added a brief paragraph in the discussion section after having spoken with colleagues that work in the field and sighting of recent literature, see line 755ff:

The progress of artificial intelligence and machine learning and its potential for automatization of qualitative judgments of texts

Since this study was conducted, machine learning and artificial intelligence applications that digest text for specific purposes have accelerated in their development and application. While problems like hallucinations persist and high accuracy at scale required for most academic work has not been achieved yet, programs become progressively better in text-related tasks, especially extraction of information from text (54–57). Despite the persisting challenges, researchers see an increasing role of AI in research and for reviews in particular (58). Future studies could compare results across human and electronic raters to assess and monitor if and when these tools will be of the needed quality to apply for contextual classification tasks.

Reviewer #2: This is a really interesting and important paper. It is well written and makes some useful arguments. I am happy to recommend the paper be published if the authors can attend to some minor amendments and suggestions below:

Methods: I agree with the authors on pg 8 lines 178-184 that the information about the systematic review and PRISMA figure can be removed. Instead the authors could provide a short description of the original study and reference it (since it has been published)

We welcome the suggestion to replace the statement about PRISMA with a summary, and have replaced the section with the following text as an introduction to the methods:

Line 220ff:

To assess the impact of group discussion on the quality of categorical classification of text, we drew on a published dataset of decision-making processes for conservation management (25). The study sampled the peer-reviewed literature with several search strings to find published texts on conservation management decisions that were derived through a prioritization process. The first author coded all texts for several variables and respective categories, requiring some degree of interpretation, and provided quantitative summaries of these categories as results. The group discussions were used as a means to validate the quality of the codes, with the author (in the following referred to as “main rater”) facilitating the experiment and comparing their own ratings and arguments with a group after independent parallel coding.

Some suggestions:

1. On pg 10, subsection titled Workflow – the authors refer to 25 studies used to illustrate their argument as ‘examples’ – this is a bit confusing – I would suggest they use the terms 25 studies, and coding on 23 categories (were there sub-codes?) and amend all references to ‘examples’ as used above to studies or publications (to make it absolutely clear what they are referring to).

Subsequently the authors refer to ‘case study’ and it is not clear what they are referring to – an ‘example’ or results of a round of coding?

Many thanks for highlighting confusing descriptions. We have changed the text throughout accordingly.

2. Perhaps the author/s could mention who the main rater was – and their expertise that made them the main rater earlier in the paper (it comes very late in the discussion). It would also be useful to mention whether any training was given to the additional raters before the process and whether they all had domain expertise (albeit to varying levels).Although on pg 27 it is mentioned that in case the studies being coded are very heterogeneous, training of coders can be difficult so as to improve coding agreement – nevertheless some initial training needs to have been provided and some mention of that would be useful.

Many thanks for these suggestions to describe the process more clearly.

We have included some text in the new introduction to the methods to explain the role of the main rater in line 220 ff, as described in response to the reviewer comment in line 166 in this document.

Reference to training is also given in line 245 (we added “To provide some introduction and training,” to make it more clear that the pilot study aimed at providing training), we also provided information on the expertise of the students in line 238 (“All raters were students of environmental sciences with a focus on conservation.”) and pulled up text from line 262ff to line 239ff have all information regarding expertise and training of the group in one paragraph.

3. Pg 10: line 245 – perhaps the authors would like to clarify whether each time different 3 -5 studies were used in each round of subsequent coding – [it is a little confusing whether all studies were coded only once by coders (in different rounds) or some studies were coded twice or more in different rounds.

Thank you for pointing out that the text is not clear enough. We have added the following and hope this describes the process in more clarity:

Line 252ff: The work was split up into several sessions with a duration of a few hours each to allow raters to remember the details of the rated publications for the group discussion, before rating the next set of publications. Six rounds of coding were needed to work through all publications , each consisting of individual rating of text followed by a group discussion.

4. From the description it appears to me that the group discussion took place only after all the rounds of coding were completed – If that was the case, I am not sure what was the point of doing the coding piecemeal in rounds? It is a bit unclear.

Please see the response to the previous point, we hope the edits have made it clearer that the consecutive rounds enabled raters to remember the studies and have sessions that were not too long.

5. Were the authors able to conclude whether there is an optimum number of additional coders – (marginal benefit of additional coders and at what point does it become negative)? It might also be worth speculating whether the marginal cost of an additional coder (time, effort and funds) is worth the marginal benefit?

That is a very good question and would be likely to be easier to answer if more studies would exist that test for the agreement rates between different additional raters. Very likely, individual characteristics of each person matter, such as prolonged concentration span and attention to detail, but in some cases also knowledge of specific terminology and how it relates to the classification codes at hand. For example, if only one student can be hired as a comparison, it might come down to bad luck if this student might be a poor performer, but with a larger group, it becomes more likely that there is a range of skill and dedication present in the group. As can be seen in figures 4 and 5, having only one additional rater likely will not catch as many possible disagreements as 2,3 or more additional raters. I see this as you suggest as well, as a trade-off between feasibility and certainty. As the study shows, a test with additional raters can lead to the insight that some categories should be rather collapsed as there are high disagreement rates among independent raters. In the case of this study, the produced codes of the main author passed the test of scrutiny, however, I would not assume that is necessarily always the case. I have added some text in line 751ff:

In light of our results, we highly recommend including a targeted inter-rater-reliability test in any classifi

---

## [Editor Report · Decision Letter 1]

Mar 23 2025

Dear Dr. Beher,

Thank you for submitting your manuscript to PLOS ONE. After careful consideration, we feel that it has merit but does not fully meet PLOS ONE’s publication criteria as it currently stands. Therefore, we invite you to submit a revised version of the manuscript that addresses the points raised during the review process.

I appreciate your detailed responses to the reviewers' comments. I believe that your manuscript is nearly suitable for publication, but I noticed some minor things -- mostly editorial -- that you should address. They are listed in the Additional Editor Comments section below. If you resolve these, I should be able to render an acceptance decision quickly.

We look forward to receiving your revised manuscript.

Kind regards,

Frank H. Koch, PhD

Academic Editor

PLOS ONE

Journal Requirements:

**Additional Editor Comments:**

Line 43: I suggest "meta-analyses" instead of the singular "meta-analysis".

Lines 65-67: The end of the sentence reads awkwardly. I suggest the following rewrite: "...that we found in our Web of Science search reported completion of any reliability checks or even stated the general importance of doing so."

Line 72: Insert "that" between "coding" and "requires".

Line 78: Rewrite: "...interpreting the context in which a word is used and making..."

Line 96: Delete second appearance of "against" on this line.

Line 98: Insert "the" between "in" and "form".

Lines 106-108: I suggest moving "therefore" to the beginning of the sentence, i.e., "Therefore, a thorough, rigorous process..."

Line 164: I suggest inserting "by the raters" between "reflection" and "on".

Line 165: I suspect you meant "beliefs" instead of "believes".

Line 257 and elsewhere: When you mention kappa in the results, are you referring to Fleiss' kappa, Light's kappa, or something else (i.e., an agreement measure that allows for more than two raters)? Please clarify.

Line 259: Replace "does" with "do".

Line 261: "Fleiss' kappa" instead of "Fleiss's kappa".

Line 261-262: "Krippendorff's alpha".

Line 262: "Scott's pi".

Line 282: replace semi-colon with a colon.

Line 329: Which kappa?

Line 329: "Krippendorff's".

Line 330: Replace "general" with "generally".

Line 331: Delete "in general".

Line 339: Replace "show" with "showed".

Line 341: Replace "pass" with "passed".

Lines 346-347: Which kappa?

Line 373, 378, and 382: Which kappa?

Line 378: Which kappa?

Line 433: Which kappa?

Line 434: Replace "doesn't" with "does not".

Line 435: Replace "won't" with "will not".

Line 438: Replace "allows" with "makes it possible".

Lines 442-443: Which kappa?

Lines 453-454: Rewrite as "disagreement due to ambiguity".

Line 575: Suggest replacing "used" with "posed".

Line 586: Suggest inserting "careful" or "close" after "pay".

Line 587: Delete "to look out for".

Line 685 ("hallucinations"): I'm not sure what you mean here. What is a hallucination with respect to an AI algorithm / model?

---

## [Author Response · Author response to Decision Letter 2]

23 May 2025

Dear Dr. Koch,

Please find the list of suggested edits from the last review and our responses, in blue, including line numbers, below.

I hope the manuscript meets now the standard of PLOS ONE and can be accepted for publication. If there is anything else, I will be of course more than happy to address any new or other issues.

Kind regards,

Jutta Beher

Additional Editor Comments:

Line 43: I suggest "meta-analyses" instead of the singular "meta-analysis".

Thank you for the suggestion, the change is made.

Lines 65-67: The end of the sentence reads awkwardly. I suggest the following rewrite: "...that we found in our Web of Science search reported completion of any reliability checks or even stated the general importance of doing so."

Thank you for the suggestion, the change is made.

Line 72: Insert "that" between "coding" and "requires".

Thank you for the suggestion, the edit is made.

Line 78: Rewrite: "...interpreting the context in which a word is used and making..."

Thank you for the suggestion, the edit is made.

Line 96: Delete second appearance of "against" on this line.

Thank you for the suggestion, the edit is made

Line 98: Insert "the" between "in" and "form".

Thank you for the suggestion, the edit is made.

Lines 106-108: I suggest moving "therefore" to the beginning of the sentence, i.e., "Therefore, a thorough, rigorous process..."

Thank you for the suggestion, the edit is made.

Line 164: I suggest inserting "by the raters" between "reflection" and "on".

Thank you for the suggestion, the edit is made.

Line 165: I suspect you meant "beliefs" instead of "believes".

Thank you for the correction, the edit is made.

Line 257 and elsewhere: When you mention kappa in the results, are you referring to Fleiss' kappa, Light's kappa, or something else (i.e., an agreement measure that allows for more than two raters)? Please clarify.

Thank you for raising the unclear referencing to the metric. I have edited the sentence in line 257 to: Fleiss’ kappa (in the following referred to as “kappa”),

To clarify that here and in the following we refer to Fleiss’ kappa whenever kappa is mentioned.

Line 259: Replace "does" with "do".

Thank you for the suggestion, the edit is now in line 260

Line 261: "Fleiss' kappa" instead of "Fleiss's kappa".

Thank you for the correction, the edit can be seen in line 262.

Line 261-262: "Krippendorff's alpha".

Thank you for the correction, the edit can be seen in line 262.

Line 262: "Scott's pi".

Thank you for the correction, the edit can be seen in line 262.

Line 282: replace semi-colon with a colon.

Thank you for the suggestion, the edit can be seen in line 289.

Line 329: Which kappa?

Please see our edit regarding your comment further up, where we indicate that we refer to Fleiss’ kappa throughout. Please let us know if you prefer to insert “Fleiss’” in front of any mention of kappa instead.

Line 329: "Krippendorff's".

Thank you for the correction, the edit can be seen in line 330.

Line 330: Replace "general" with "generally".

Thank you for the suggestion, the edit can be seen in line 331.

Line 331: Delete "in general".

Thank you for the suggestion, the edit can be seen in line 332.

Line 339: Replace "show" with "showed".

Thank you for the correction, the edit can be seen in line 341.

Line 341: Replace "pass" with "passed".

Thank you for the correction, the edit can be seen in line 342.

Lines 346-347: Which kappa?

Line 373, 378, and 382: Which kappa?

Line 378: Which kappa?

Line 433: Which kappa?

Please see our edit regarding your comment further up, where we indicate that we refer to Fleiss’ kappa throughout. Please let us know if you prefer to insert “Fleiss’” in front of any mention of kappa instead.

Line 434: Replace "doesn't" with "does not".

Thank you for the suggestion, the edit can be seen in line 435.

Line 435: Replace "won't" with "will not".

Thank you for the suggestion, the edit can be seen in line 436.

Line 438: Replace "allows" with "makes it possible".

Thank you for the suggestion, the edit can be seen in line 439.

Lines 442-443: Which kappa?

Please see our edit regarding your comment further up, where we indicate that we refer to Fleiss’ kappa throughout. Please let us know if you prefer to insert “Fleiss’” in front of any mention of kappa instead.

Lines 453-454: Rewrite as "disagreement due to ambiguity".

Thank you for the suggestion, the edit can be seen in line 455.

Line 575: Suggest replacing "used" with "posed".

Thank you for the suggestion, the edit can be seen in line 577.

Line 586: Suggest inserting "careful" or "close" after "pay".

Thank you for the suggestion, the edit can be seen in line 588.

Line 587: Delete "to look out for".

Thank you for the suggestion, the edit can be seen in line 589.

Line 685 ("hallucinations"): I'm not sure what you mean here. What is a hallucination with respect to an AI algorithm / model?

Thank you for the question. “Hallucinations” is a term that describes any errors made by generative AI. This term is frequently used in AI related publications, some chose more blunt terms, including “bullshit”. The term “hallucination” is also used (and described) in several of the citations I refer to. I have added brief explainer in line 687, please let me know if that is not sufficient:

However, generative AI and chatbots make mistakes, too, and can for example suggest citations that do not exist or write code that does not work. Such errors are often referred to as “hallucinations”.

---

## [Editor Report · Decision Letter 2]

Group discussions improve reliability and validity of rated categories based on qualitative data from systematic review

PONE-D-24-38272R2

Dear Dr. Beher,

We’re pleased to inform you that your manuscript has been judged scientifically suitable for publication and will be formally accepted for publication once it meets all outstanding technical requirements.

Kind regards,

Frank H. Koch, PhD

Academic Editor

PLOS ONE

Additional Editor Comments (optional):

Thank you for completing a final round of edits.
---

## [Editor Report · Acceptance letter]

PONE-D-24-38272R2

PLOS ONE

Dear Dr. Beher,

I'm pleased to inform you that your manuscript has been deemed suitable for publication in PLOS ONE. Congratulations! Your manuscript is now being handed over to our production team.

Kind regards,

on behalf of

Dr. Frank H. Koch

Academic Editor

PLOS ONE